# An Exploration of Motivation for Disaster Engagement and Its Related Factors among Undergraduate Nursing Students in Taiwan

**DOI:** 10.3390/ijerph17103542

**Published:** 2020-05-19

**Authors:** Shwu-Ru Liou, Hsiu-Chen Liu, Chun-Chih Lin, Hsiu-Min Tsai, Ching-Yu Cheng

**Affiliations:** 1College of Nursing, Chang Gung University of Science and Technology, No. 2, Sec. W., Jiapu Rd., Puzi City, Chiayi County 613, Taiwan; slwu@mail.cgust.edu.tw (S.-R.L.); lhc@mail.cgust.edu.tw (H.-C.L.); cclin01@mail.cgust.edu.tw (C.-C.L.); 2Department of Nursing, Chiayi Chang Gung Memorial Hospital, No.6, Sec. W., Jiapu Rd., Puzi City, Chiayi County 613, Taiwan; 3College of Nursing, Chang Gung University of Science and Technology, No. 261, Wenhua 1st Rd., Guishan Dist., Taoyuan City 33303, Taiwan; hmtsai@mail.cgust.edu.tw

**Keywords:** disaster, competence, stress, motivation, nursing student

## Abstract

The purpose of the study was to explore the levels of and relationships between disaster nursing competence, anticipatory disaster stress, and the motivation for disaster engagement among undergraduate nursing students in Taiwan. A cross-sectional research design was applied. Using convenience sampling, 90 nursing students participated with an 86.54% response rate. The Disaster Core Competencies Questionnaire, Anticipatory Disaster Stress Questionnaire, and Motivation for Disaster Engagement Questionnaire were used to collect data. The Pearson correlation and ANOVA were used to analyze the data. Results showed that students’ level of disaster nursing competence was low, anticipatory disaster stress was not high, and motivation for disaster engagement was high. Motivation for disaster engagement was positively correlated with anticipatory disaster stress. Students who were more willing to participate in disaster management had a higher level of anticipatory disaster stress and motivation for disaster engagement. It is suggested that healthcare institutions and schools should work together to design disaster education plans using innovative teaching/learning strategies to increase students’ willingness and motivation for disaster engagement.

## 1. Introduction

Among all continents in the world, Asia was the one most impacted by disasters in terms of the number of deaths as well as disaster events [1]. Regarding disaster mortality, Taiwan was listed in the top 10 Asian countries [2]. The consequences of disasters cause many long-term serious environmental disruptions and mental impairment to the survivors in the community [3,4,5]. Therefore, nurses, who are the largest group of healthcare professionals, are expected to quickly provide care to wounded people when the disaster occurs to reduce the disruptive impacts of disasters on community residents [6,7]. Recently, a growing consensus has emerged about the importance of furnishing all nurses with a minimum level of knowledge and skills that will allow them to face the complexities of disasters [8,9]. Even though increasing nurses’ level of disaster nursing competence, skills, and experience has been emphasized, nurses are considered unprepared and inadequate for providing disaster management and care [6,10,11].

More significantly, studies reveal an alarming issue that nurses have less motivation to report to work although they may believe they are responsible for working during disaster situations [12,13]. Studies disclose that high levels of psychological stress are frequently reported in emergency responders during disaster events [14,15], which may be the primary reason affecting nurses’ motivation to report to disaster or casualty sites.

The occurrence of disaster events, particularly when the events cause many casualties or the disasters last for an extended period of time, is often an indicator that the capacity of the nursing workforce in a community or health institution is being overwhelmed. Researchers have proposed to deliberate more comprehensively on the role of nursing students in supporting disaster responses. Researchers predict that nursing students are a potential group that can increase the volume of the nursing workforce and contribute to disaster relief during or after disaster events when nursing shortages are a problem [16,17]. Therefore, nursing students’ disaster nursing competence is of concern academically and clinically [17,18,19,20,21].

Both nursing students and registered nurses possess important roles in responding to disaster events. Nevertheless, both of them were found to be not well prepared in disaster management [7,22]. Nursing students are future nurses. Therefore, healthcare administrators and nursing faculty are increasingly concerned about nursing students’ preparedness for disaster events before they graduate [18,19,20,21]. The literature discusses disaster-related issues among registered nurses but there is scarce data describing the individual level of disaster nursing competence, anticipatory disaster stress, and motivation for disaster engagement, and their relationships among nursing students.

Inadequate competence and high stress may impact nursing students’ motivation to engage in disaster events once they are employed as working nurses. The purpose of the study was to explore the relationships between disaster nursing competence, anticipatory disaster stress, and motivation for disaster engagement among nursing students in Taiwan. The research was guided by the following questions: (a) What are the degrees of disaster nursing competence, anticipatory disaster stress, and motivation for disaster engagement among nursing students? (b) What are the relationships between disaster nursing competence, anticipatory disaster stress, and motivation for disaster engagement? (c) What are the differences between disaster nursing competence, anticipatory disaster stress, and motivation for disaster engagement by demographic variables? The findings of the study can expand our knowledge about nursing students’ status on disaster nursing competence, anticipatory disaster stress, and motivation for disaster engagement and, therefore, facilitate nursing school faculties and healthcare managers to design and implement learning/training programs to prepare a competent workforce.

## 2. Background

### 2.1. Disaster Nursing Competence

Competence has been defined inconsistently in literature; however, many definitions are essentially similar. The National Council for State Boards of Nursing [23] defines nursing competency as the “application of knowledge, interpersonal decision-making, and psychomotor skills expected for the practice role within the context of public health.” The International Council of Nurses (ICN) defines competence as “a level of performance demonstrating the effective application of knowledge, skill, and judgment” [24]. The concept of disaster nursing concentrates on the systematic and flexible utilization of knowledge, skills, and activities to provide holistic care and decrease health hazards for populations in collaboration with other professional fields during all phases of a disaster [25,26]. To achieve the goals of disaster nursing, all nurses must have core competencies. There is a paucity of frameworks constructed for disaster core competencies [27]. The most common core competencies identified as essential can be found in the “Framework of Disaster Nursing Competencies” developed as a collaboration between the ICN and the World Health Organization (WHO). The framework includes 10 domains and 130 core competencies [28]. The ICN proposes that disaster-competent nurses demonstrate not only the basic nursing competencies but also the competencies necessary when working in disaster situations [28].

Researchers propose that hospital nurses’ disaster competencies need to be improved even though they may have been trained in dealing with some emergency events in hospital settings [22,29]. However, nurses are found to not be confident in their abilities to respond to disaster events [30,31]. Similarly, nurses report their self-perceived disaster nursing competencies are grossly inadequate in dealing with complex critical situations in the communities they serve. Such inadequate competencies are specified as skills to care for injured people, knowledge about the disaster, and specific interventions related to a disastrous situation [21,32]. In comparison, the disaster competence of nursing students, who will be our future nurses, has not been fully studied. One study had found that nursing students were not well-prepared to respond in a disaster setting [33].

### 2.2. Anticipatory Disaster Stress

In this study, stress is defined as “a relationship between the person and the environment that is appraised by the person as relevant to his or her well-being and in which the person’s resources are taxed or exceeded” [34] (p. 150). Disaster stress therefore means that individuals are inclined to exhibit psychological problems and distress when involved in disaster events [14], especially when the required tasks exceed their abilities. The sources of stress may come from the disaster itself, the environment of the situation, and the safety of the nurses themselves and that of others during the disaster event. Individuals worry about the recurrence of the disaster event, their workload and challenges in the disaster field, working in a chaotic situation with scarce resources, availability of protective equipment, individual safety, family concerns, fear of unknown, fear of not being able to perform, and working for an unknown period of time [35,36,37,38]. Specifically, nurses with previous disaster response experience expressed that they had a feeling of fear at the moment when waiting for victims to arrive during a disaster situation they described as very emotional and extremely tense [37,39]. All of these concerns may affect nurses’ motivation to attend or remain at the situation site.

### 2.3. Motivation for Disaster Engagement

Motivation is important in human behavior and is the force that causes movement in humans [40]. Motivation explains the start, direction, and perseverance of behavior among individuals involved with adding value to the goals, perceived competence, causal attributions, and emotional reactions [41]. Research studies [40] indicated that individuals with high job motivation showed greater commitment to their duty. However, engagement in disaster relief, and even motivation to engage in it, is a challenge for nurses. Studies reported a wide range (36%–80%) in rates of hospital healthcare providers who showed the intent to report to work during a disaster event. Their intention depended on the types of disasters and factors that might threaten individuals’ personal safety [42,43,44]. In addition, nurses’ perceived disaster nursing competencies were commonly insufficient [21,30]. Nurses were also considered to not take basic actions to prepare themselves for disaster management personally or professionally [30,31]. They also proposed that people had different motives; therefore, before acting on the motivation, it was necessary to know the individuals’ personal characteristics and their driving forces [40].

Exploring and promoting the motivation for disaster engagement among nursing students is imperative. Researchers proposed that motivating nursing students to engage in disaster rescue activities could promote better readiness in the future when encountering disasters as nurses [45]. However, a low motivation to engage in disaster-related activities also existed among nursing students [33]. Although nursing students expressed a high willingness to be involved in disaster responses, they were found to not actively engage in disaster activities [33,46]. Shannon [46] designed a project to improve nursing students’ engagement in community disaster preparedness; however, the author concluded that the engagement of students in disaster management efforts remains a challenge in nursing education. Interventional studies involving animals or humans and other studies requiring ethical approval must list the authority that provided approval and the corresponding ethical approval code.

### 2.4. Relationships between Disaster Nursing Competence, Anticipatory Disaster Stress, and Motivation for Disaster Engagement

Motivation has a component of choice, intention, or willingness [47], while competence is the feeling of confidence that can drive individuals to complete missions and do them well [48]. Deci and Ryan [49] pointed out that individuals’ motivation for engagement behaviors and maintenance of the motivation over a period of time are influenced by the individuals’ competence. That is, individuals’ perceived competence when performing actions is important since individuals who feel more competent are more confident and have higher levels of motivation to perform the expected actions. On the contrary, individuals experiencing incompetence or low confidence in their abilities may sense anxiety, depression, stress, dissatisfaction, helplessness, and lower self-esteem [50,51,52], which further affects their motivation, choice of activities, willingness to take responsibilities or challenges, and the length of taking responsibilities [52,53].

Researchers proposed that personnel involved in casualty events are prone to exhibiting psychological problems and distress [14]. Experts working in the disaster field pointed out that the nature of potential stressors, incompetence in response to disaster events, and lack of mediators associated with a disaster event may influence individuals’ commitment to attend or remain at the casualty site [30,39]. One study found that behavior motivation was affected by perceived nurse competence in managing disasters only in regard to the nurse’s willingness to assume the risk of involvement in a disaster situation [30].

Based on the literature review, we hypothesized that anticipated disaster competence was negatively correlated with anticipatory disaster stress and positively correlated with motivation for disaster engagement whereas anticipatory disaster stress was negatively correlated with motivation for disaster engagement (Figure 1).

## 3. Materials and Methods

### 3.1. Design and Sample

This study was a cross-sectional correlational design. Because there was no research studied concurrently regarding the variables we planned to measure in this study, the sample size was calculated based on an estimation so that the effect size for correlation between measured variables was at least median (r = 0.3) [54]. Using G*Power version 3.0 [55] with a power of 0.80, two-tailed test, and α level of 0.05, the estimated sample size was 82 for the study. More nursing students were contacted to participate in the study in case of a possible incompletion rate. Using convenience sampling, nursing students who were enrolled in the investigator’s serving university and who met the inclusion criteria were contacted and recruited. The inclusion criteria included students who (a) had taken and passed the medical–surgical nursing courses and clinical practices; (b) had passed or were taking obstetric nursing, pediatric nursing, psychological nursing, and community nursing courses; and (c) were willing to sign an agreement to participate in the study and complete questionnaires. In all, 104 students were contacted and responded to our invitation to participate in the study, and 90 of them returned completed questionnaires with an 86.54% response rate.

### 3.2. Ethical Consideration

We began to conduct the study after obtaining approval from the Institutional Review Board committee (IRB no.: 201601662B0). The study’s purposes, procedures, participants’ rights, and ethical considerations were explained to potential participants who were interested in the study. Signed consent was obtained before the study started. Participants were given a packet containing a set of questionnaires and a cover letter that described the participants’ rights, study purpose, risks and benefits of participation, and information about confidentiality. Participants were assured that they had the right to not fill out the entire questionnaire or answer any questions that they were not comfortable with answering. They could fill out the questionnaires at any location where they felt comfortable. All participants turned in the signed informed consent and completed questionnaires to the investigators separately by using two addressed-and-stamped envelopes.

### 3.3. Instruments

#### 3.3.1. Demographic Sheet

The demographic sheet contained questions relating to age, gender, experience with encountering disasters, planned job type after graduation, and willingness to participate in disaster management. Questions regarding the participants’ preferred education methods and preferred class schedule arrangement for disaster courses were also asked.

#### 3.3.2. Disaster Core Competencies Questionnaire (DCCQ)

We developed and constructed the DCCQ based on the ICN Framework of Disaster Nursing and its indicators to examine nursing students’ disaster nursing competence. The ICN Framework of Disaster Nursing includes 10 domains, with each domain containing several indicators for evaluating nurses’ disaster nursing competencies [28]. The DCCQ is a 26-item instrument using a five-point Likert scale scoring from 1 (do not have a clue) to 5 (know theoretically, confidence in familiarity without any supervision). A higher score indicates a higher level of disaster nursing competence. In the study, the internal consistency reliability tested using Cronbach’s alpha was 0.95. The principal component analysis for the validity showed that 62.50% of the variance of disaster nursing competence could be explained by the DCCQ.

#### 3.3.3. Anticipatory Disaster Stress Questionnaire (ADSQ)

The ADSQ was used to measure nursing students’ perception of stress when they confront disaster events. This 24-item scale was developed by the researchers based on literature reviews. The items were rated on a five-response Likert scale scoring from 1 (strongly disagree) to 5 (strongly agree). A higher score indicates that the individuals perceive greater stress while facing disaster events. In the study, the internal consistency reliability tested using Cronbach’s alpha was 0.74. The principal component analysis showed that 44.87% of the variance of the anticipatory disaster stress could be explained by the ADSQ.

#### 3.3.4. Motivation for Disaster Engagement Questionnaire (MDEQ)

The MDEQ was used to explore students’ motivation for disaster engagement. The three-item scale was developed by the researchers and rated in five-point Likert scale (ranging from 1–5). A higher score indicates a higher level of motivation to attend disaster events. In this study, the internal consistency reliability tested using Cronbach’s alpha was 0.75. Principal component analysis showed that 67.09% of the variance of the motivation in disaster engagement could be explained by the MDEQ.

### 3.4. Data Analysis

The collected data were managed and analyzed using the SPSS version 23.0 [56]. Descriptive statistics were used to understand the participants’ demographic characteristics and levels of disaster nursing competence, anticipatory disaster stress, and motivation for disaster engagement. The normality of measured variables was examined, and the results indicated that disaster nursing competence (DCCQ) and anticipatory disaster stress (ADSQ) were normally distributed, whereas motivation for disaster engagement (MDEQ) was not normally distributed. The Pearson correlation was used to examine the relationships between measured variables when the variables were normally distributed, and the Spearman correlation was used when the variables were not normally distributed. Student’s t-test or Mann–Whitney U test was used to compare measured variables by demographic variables that were two levels whereas ANOVA or Kruskal–Wallis test was used when demographic variables were more than two levels. The reliability of all scales was tested using internal consistency (Cronbach’s alpha coefficient), and validity was tested with a principal component analysis.

We did a path analysis to test relationships between measured variables using AMOS version 24.0 [57]. Because the outcome variable (motivation for disaster management) was not normally distributed, generalized least square was used to analyze the data. The Χ^2^, chi-square/degree-of-freedom ratio (Χ^2^/df), root mean square error of approximation (RMSEA), and comparative fit index (CFI) were used as indicators to determine the fit of the model to the data. The cutoff value for a good fit of the model to the data was set as *p* > 0.05 for Χ^2^, <5 for Χ^2^/df, <0.06 for RMSEA, and >0.95 for CFI [58].

## 4. Results

### 4.1. Demographic Information of the Participants

The mean age of the participants was 21 (SD = 0.62) years and most participants were females (91.1%). While 55.6% of students were in the first year of the two-year bachelor program, 44.4% were in the third year of the four-year bachelor program. It was found that 90.0% of the students would like to work as a nurse after graduation, and 36.7% planned to work in medical/surgical units, 14.4% intended to work in intensive care units and 23.3% in emergency rooms after graduation. Moreover, 94.4% of the students had not taken a disaster course before, 54.4% had not seen/experienced a disaster before, only 13.3% regarded themselves as very familiar with disaster management procedures while 42.2% were very unfamiliar or unfamiliar with disaster management procedures, and 27.8% were willing to participate in disaster management.

Most nursing students thought that their school should offer a disaster nursing course (87.8%) and agreed (48.9%) or extremely agreed (21.1%) that students should take a disaster nursing course before graduation. The top three ways the students preferred to learn disaster nursing were face-to-face class learning (87.8%), online learning (62.2%), and classroom plus practicum (50.0%). Specifically, a course combined with one-day practice (83.3%), a two-credit-hour course (77.8%), or a face-to-face plus online teaching course (75.6%) was preferred.

### 4.2. Relationship between Disaster Nursing Competence, Anticipatory Disaster Stress, and Motivation for Disaster Engagement

As shown in Table 1, the levels of disaster nursing competence (M = 71.81, SD = 15.49) were low, and anticipatory disaster stress (M = 76.90, SD = 7.33) among nursing students was moderate. The degree of motivation for disaster engagement (M = 12.03, SD = 1.24) among these students was high. Among the variables of disaster nursing competence, anticipatory disaster stress, and motivation for disaster engagement, only the motivation for disaster engagement was moderately correlated with anticipatory disaster stress (r = 0.31, *p* = 0.003).

### 4.3. Differences between Disaster Nursing Competence, Anticipatory Disaster Stress, and Motivation for Disaster Engagement by Demographic Variables

Most of the demographic variables had no statistical impact on students’ disaster nursing competence, anticipatory disaster stress, and motivation for disaster engagement. As shown in Table 2, participants who would work as a registered nurse (RN) in hospitals or healthcare-related institutions had a higher score on motivation for disaster engagement (U = 20.00, *p* = 0.02). Those who regarded themselves to be familiar with disaster management procedures (F = 15.87, *p* < 0.001) had higher scores on disaster nursing competence. Participants with higher degrees of willingness to participate in disaster management had higher scores on anticipatory disaster stress (F = 7.49, *p* < 0.001) and motivation for disaster engagement (Χ^2^ = 21.89, *p* < 0.001). Level of motivation for disaster engagement differed by variables of “nursing schools must offer a disaster nursing course” (Χ^2^ = 8.38, *p* = 0.02) and “nursing students must take that course before graduation” (Χ^2^ = 12.38, *p* = 0.01).

### 4.4. Model Test of Disaster Nursing Competence, Anticipatory Disaster Stress, and Motivation for Disaster Engagement by Demographic Variables

The original proposed model did not fit the data (Χ^2^ = 0.00, *p* could not be computed), RMSEA = 0.19, CFI = 1.00). The path coefficient between disaster nursing competence and disaster stress (β = 0.20, *p* = 0.05) and that between disaster nursing competence and motivation for disaster engagement (β = −0.04, *p* = 0.67) were not significant while the path coefficient between disaster stress and motivation for disaster engagement (β = 0.36, *p* < 0.001) was significant. After removing the path between disaster nursing competence and motivation for disaster engagement, the Χ^2^ became 0.18, *p* = 0.67, Χ^2^/df was 0.18, RMSEA was 0.00, NFI was 0.99, and CFI was 1.00. The model showed a good fit to the data (Figure 2). The association between disaster nursing competence and motivation for disaster management was not high (β = 0.20, SE = 0.05, t = 1.96, *p* = 0.05) and the association between disaster stress and motivation for disaster engagement was significant (β = 0.35, SE = 0.02, t = 3.48, *p* < 0.001).

Another path analysis was conducted by including demographic variables (i.e., job after graduation, willing to participate in disaster management, familiar with disaster management procedures, school needs to offer a disaster nursing course, and should take disaster nursing course before graduation) that were associated with disaster nursing competence, disaster stress, or motivation for disaster engagement in the model. Results showed that the model did not have a good fit to the data (Χ^2^ = 71.14, *p* < 0.001; Χ^2^/df = 3.56; CFI = 1.00; and RMSEA = 0.17). Only “willingness to participate in disaster management” was significantly associated with disaster stress (β = 0.28, *p* = 0.01) and motivation for disaster engagement (β = 0.24, *p* = 0.04). After removing demographic variables that did not have significant effects on measured variables, the Χ^2^ became 1.95, *p* = 0.38, Χ^2^/df was 0.98, RMSEA was 0.00, and CFI was 1.00. The model showed a good fit to the data (Figure 3). Disaster nursing competence was not significantly associated with disaster stress (β = 0.15, SE = 0.05, t = 1.53, *p* = 0.13). “Willingness to participate in disaster management” was significantly associated with disaster stress (β = 0.39, SE = 0.93, t = 3.92, *p* < 0.001) and motivation for disaster management (β = 0.34, SE = 0.16, t = 3.37, *p* < 0.001). The association between disaster stress and motivation for disaster engagement was significant (β = 0.21, SE = 0.02, t = 2.05, *p* = 0.04) as well. A total of 22% of the variation of the motivation for disaster engagement could be explained.

## 5. Discussion

The purpose of the study was to explore the levels of and relationships between disaster nursing competence, anticipatory disaster stress, and motivation for disaster engagement among nursing students in Taiwan. Additionally, the factors associated with the measured variables were identified. Among the measured variables in our study, only anticipatory disaster stress was correlated with the motivation for disaster engagement. Most of the demographic variables did not have a significant influence on students’ disaster nursing competence, anticipatory disaster stress, or motivation for disaster engagement.

The level of disaster nursing competence among nursing students was low in this study. This finding of low competence is congruent with findings in previous studies [26,33], but is not in line with the result from another study on nursing students [22]. However, the investigators of the latter study considered and commented that nursing students in the study might have over-estimated their disaster nursing abilities [22]. Nevertheless, we found that students who were more familiar with disaster management procedures had higher levels of disaster nursing competence. Therefore, education or training for disaster management might improve nursing students’ disaster competence.

In contrast from the experiences of practicing nurses, the level of anticipatory disaster stress among nursing students in our study was not high. Regarding disaster stress, nurses were found to be prone to exhibiting psychological problems and distress when participating in events with casualties [14]. Since nursing students did not have any experience in disaster field rescue, it might be hard for them to picture the stress they might encounter when they come across a disaster. In addition, the nursing students who were surveyed were in the millennial generation characterized by confidence, optimism, and high self-esteem [59,60]. Therefore, nursing students may have underestimated their disaster stress level.

In contrast to the high level of motivation for disaster engagement found in our study, Japanese nursing students did not have a high motivation for participating in disaster relief activities in either domestic areas or foreign countries as nurses in the future [45]. Nevertheless, only 27.8% of nursing students in our study expressed that they were willing to participate in community disaster relief. Unsurprisingly, we found that nursing students who had a high level of willingness to participate in disaster management also had a higher level of motivation for disaster engagement. From these findings, we presumed that the willingness may play a vital role in motivation for disaster engagement. Broussard and Garrison broadly defined motivation as “the attribute that moves us to do or not to do something” [61] (p. 106). Willingness is defined as “the quality or state of being prepared or ready to do something” [62] (p. 80). Based on these definitions, students’ willingness to participate in disaster management might be higher if students could clearly comprehend why and what to do in their tasks in disaster events, which might further motivate students to get involved in the expected task.

In addition to the nursing students in our study who were less willing to participate in disaster management, previous studies also found that nursing students were not interested and actively engaged in disaster actions [33,46]. In contrast, one study found that the willingness to respond to a disaster among junior and senior nursing students varied by the type of disaster [43]. Both nursing students in our study as well as health professionals in other studies had low intentions to respond to disaster relief. Researchers [38,63] pointed out that health professionals were less willing to respond to incidents involving potential exposure to harmful agents such as chemical, radiological, or biological agents. Nurses also were less willing to report to work during disaster situations or attend disaster relief because they were concerned about unforeseeable circumstances that might happen during a disaster [12,13,14,35,36]. The main obstacles to willingness to participate in disaster management included fear and worry about their own safety, the availability of protective equipment, and their education and training in disaster preparedness [38].

In our study, disaster nursing competence was not correlated with anticipatory disaster stress—a result that was different from previous studies that had explored the relationship between nursing competence and stress. One study revealed that nursing students’ nursing competence was negatively correlated with stress during practice [64] and the other study found a negative correlation between nurses’ competence and burnout [65], which could be predicted by occupational stress [66]. This finding was also in contrast with Bandura’ theory that people with low confidence levels in competence are prone to experience stress, depression, anxiety, and helplessness [52]. The characteristics of the millennial generation might be able to explain this deviation. The surveyed nursing students were born around 1996 and are considered as part of the millennial generation [60]. Millennials are generally confident, yet have low levels of self-competence but high levels of self-evaluation. Their confidence comes from their trust and optimism, high self-esteem, and assertiveness when compared with prior generations at a similar age [59,60,67]. These characteristics of the millennials may lead nursing students to not worry about managing future disaster events because they feel confident to do the job and do it well although they may not be competent enough on disaster management.

In contrast to the previous theory that postulates that individuals’ competence can influence their motivation in engagement behaviors [41,49], we found in our study that competence was not correlated with motivation for disaster engagement among nursing students. Nevertheless, Baack and Alfred [30] found that only under the circumstance of nurses’ willingness to take on risky tasks in disaster situations, their perceived competence to manage disasters influenced their behavior motivation. This might be the reason that disaster nursing competence was not correlated with motivation in engagement behaviors in our study.

Most nursing students in the current study would like to work as registered nurses in the emergency room or intensive care unit of a hospital after graduation. Meaningfully, most nursing students expressed the necessity of a disaster course before graduating from college. These findings were somewhat congruent with prior studies in which most nursing students thought that nurses in emergency rooms or critical care units should play a significant role when disaster events occur and that nursing curriculums should include disaster nursing [18,26]. Nursing school educators have also supported a required disaster nursing education or training component for nursing students before they graduate [18,19,20,21]. We found in this study that students’ preferred method of teaching/learning disaster nursing was school lessons combined with disaster practice. Similar to results found in Ozpulat and Kabasakal’s study [18], students preferred to have a two-hour class lecture per week or classroom teaching combined with a one-day workshop; however, this finding was different from Landry and Stockton’s study [68] that lectures and videos were the less preferred method for disaster learning.

Our study has some limitations. Since the participants of this study were nursing students in the authors’ serving university in Taiwan, the findings may not be generalizable to nursing students in other settings or other countries. Because the study used a cross-sectional design, the findings cannot infer causality between variables. However, the study furnished a fundamental exploration about disaster engagement-related issues among nursing students, which has been urged by healthcare administrators and nursing faculty to probe [18,19,20,21]. Further studies are recommended to include nursing students at least in a variety of other nursing schools in Taiwan to increase the power of generalizability of the findings. Triangulation could be another study design for future research because it can provide in-depth information to understand the meaning of the measured variables among nursing students.

## 6. Conclusions

Natural or man-made disasters commonly occur and often exceed the local and regional capacity to respond. The consequences of disasters can cause a extensive human morbidity and mortality, economic losses, and serious environmental disruption. The impacts of disasters or catastrophes on human health or health behaviors may last long, especially for people with disabilities and vulnerable adolescents [69,70]. Both nursing students and registered nurses possess important roles in responding to disaster events. To decrease the extent of damage during disasters, it is imperative to promote disaster nursing competence and an authentic motivation for disaster engagement as well as to reduce anticipatory stress among nursing students. It is suggested that fostering students with a motivation for engagement in disaster rescue activities before graduation can promote them to have a better readiness in the future when encountering disasters as nurses [45]. We found in the study that nursing students did not have a high level of disaster nursing competence and anticipatory disaster stress, but their motivation for disaster engagement was high. Level of perceived anticipatory disaster stress was moderately and positively correlated with motivation for disaster engagement. In addition, students who had higher level of willingness to participate in disaster management had a higher level of anticipatory disaster stress and motivation for disaster engagement. Based on these results, we suggest that schools should design disaster-related courses and apply innovative teaching/learning strategies in their courses to increase students’ leaning interests in disaster nursing. Comprehension of disaster nursing may increase students’ willingness to participate in disaster management. While designing the disaster courses, disaster nursing competence standards proposed by the ICN and WHO can be referenced. Schools can work with healthcare institutions and local government departments to provide students with opportunities to practice disaster care.

## Figures and Tables

**Figure 1 ijerph-17-03542-f001:**
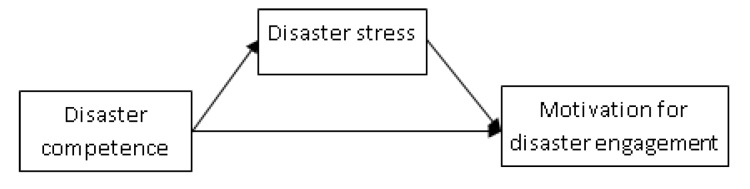
Conceptual model of the relationships between disaster stress, disaster nursing competence, and motivation for disaster engagement.

**Figure 2 ijerph-17-03542-f002:**
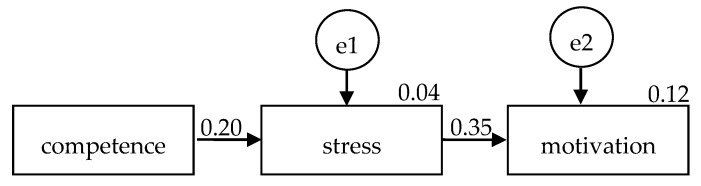
Relationships between disaster nursing competence, disaster stress, and motivation for disaster engagement.

**Figure 3 ijerph-17-03542-f003:**
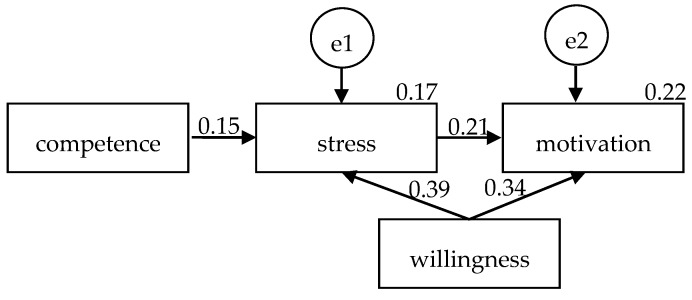
Parameter estimates of willingness to participate in disaster management, disaster nursing competence, disaster stress, and motivation for disaster engagement.

**Table 1 ijerph-17-03542-t001:** Relationships between disaster nursing competence, anticipatory disaster stress, and motivation for disaster engagement.

Variable	DCCQ	ADSQ	MDEQ
DCCQ	1.00		
ADSQ	0.20	1.00	
MDEQ	−0.10	0.31 *	1.00
M ± SD	71.81 ± 15.94	76.90 ± 7.33	12.03 ± 1.24

The Disaster Core Competencies Questionnaire (DCCQ) was used to measure disaster nursing competence, while the Anticipatory Disaster Stress Questionnaire (ADSQ) was used for anticipatory disaster stress, and the Motivation for Disaster Engagement Questionnaire (MDEQ) was used for disaster engagement. * *p* < 0.05.

**Table 2 ijerph-17-03542-t002:** Differences on the disaster nursing competence, anticipatory disaster stress, and motivation for disaster engagement by demographic variables.

Variable	Level	*n*	DCCQ		ADSQ		MDEQ	
			M ± SD	F or t	M ± SD	F or t	Mdn	Χ^2^ or U
Gender	Female	82	72.73 ± 16.41	−0.15	76.70 ± 7.52	−0.85	12.00	309.00
	Male	8	73.63 ± 10.64		79.00 ± 4.81		12.00	
Grade	First year in 2-year school	50	74.42 ± 16.47	1.07	76.38 ± 6.77	−0.75	12.00	990.50
	Third year in 4-year school	40	70.80 ± 15.21		77.55 ± 8.02		12.00	
Job after graduation	RN in hospital or healthcare-related institutions	81	73.04 ± 15.43	0.40	77.30 ± 7.44	1.55	12.00	20.00 *
	Non nursing job	9	70.78 ± 20.95		73.33 ± 5.41		11.00	
Unit plans to work in	Intensive care unit	13	79.85 ± 15.19	1.27	79.00 ± 9.14	1.18	12.00	1.20
	Emergency room	21	69.24 ± 14.09		75.38 ± 7.10		12.00	
	Medical/surgical	33	73.24 ± 17.13		78.00 ± 6.43		12.00	
	Others	23	71.48 ± 15.83		75.52 ± 7.58		12.00	
Saw/experienced disaster events before	Yes	41	70.88 ± 15.60	−1.05	76.88 ± 8.80	−0.03	12.00	969.00
	No	49	74.43 ± 16.19		76.92 ± 5.93		12.00	
Willing to participate in disaster management	1. Not at all	4	55.00 ± 21.95	2.28	65.25 ± 0.50	7.49 ** 1 < 2,3 < 4	11.50	21.89 ** 1,2,3 < 4
	2. Slightly	21	71.43 ± 17.41		75.57 ± 5.80		12.00	
	3. Moderately	46	75.48 ± 14.98		76.61 ± 6.61		12.00	
	4. Very much	19	71.63 ± 13.60		81.53 ± 8.02		13.00	
Familiar with disaster management procedures	1. Very unfamiliar	10	50.60 ± 16.23	15.87 ** 1 < 2,3 < 4	74.40 ± 8.53	0.62	12.00	2.36
	2. Unfamiliar	28	70.32 ± 10.33		77.39 ± 8.26		12.00	
	3. Neutral	40	75.68 ± 12.47		76.70 ± 6.82		12.00	
	4. Familiar	12	87.58 ± 17.13		78.50 ± 5.82		12.00	
School needs to offer a disaster nursing course	1. Neutral	11	76.82 ± 14.36	0.90	75.27 ± 6.39	2.25	12.00	8.38 * 2 < 3
	2. Need to offer	51	73.59 ± 16.00		75.94 ± 6.62		12.00	
	3. Extremely need	28	69.82 ± 16.42		79.29 ± 8.50		12.00	
Should take disaster nursing course before graduation	1. Disagree	2	74.00 ± 1.41	0.24	65.50 ± 0.71	2.48	10.50	12.38 * 2 < 4
	2. Neutral	25	70.56 ± 16.89		75.60 ± 7.07		12.00	
	3. Agree	44	73.43 ± 15.64		77.32 ± 6.25		12.00	
	4. Extremely agree	19	74.21 ± 16.74		78.84 ± 9.21		12.00	

The DCCQ was used to measure disaster nursing competence, while the ADSQ was for anticipatory disaster stress, and the MDEQ was for disaster engagement. RN: registered nurse. Mdn: median. Mann–Whitney U test (U) was used for two-level demographic variables. Kruskal–Wallis test (Χ^2^) and post hoc test with Bonferroni adjustment were used for demographic variables with more than two levels. * *p* < 0.05, ** *p* < 0.001.

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
