# Peer review of "An Exploration of Motivation for Disaster Engagement and Its Related Factors among Undergraduate Nursing Students in Taiwan"

_ijerph, 2020, doi:10.3390/ijerph17103542_

Round 1

Reviewer 1 Report

Dear Authors,
It has been a pleasure to review this work. I only find that they must modify the keywords, since those in the paper are not MeSH descriptors. This should be corrected.

Best regards

Author Response

As suggested, we have revised the keywords to MeSH terms. Thank you.

Please see page 1, line 29.

Reviewer 2 Report

The article is interesting and well presented.

I'd like to see the inclusion of an explicit linkage with the theme covered by the IJERPH - that is, both in the introduction and more in depth in the conclusion, a definition of disaster as having consequences on the environment and health, being natural disaster, accident, man made disaster. This connection can be expanded stating that a modified behavior of nurses can contribute to reduce the negative consequences on environment and health. Otherwise your paper will be purely focused on education.

Author Response

Thank you for your review of our manuscript and your willingness to allow us to revise and resubmit the manuscript. Please see below for our responses to your comments. We hope that we have addressed your major concerns.

Comment: The article is interesting and well presented.

I'd like to see the inclusion of an explicit linkage with the theme covered by the IJERPH - that is, both in the introduction and more in depth in the conclusion, a definition of disaster as having consequences on the environment and health, being natural disaster, accident, man made disaster. This connection can be expanded stating that a modified behavior of nurses can contribute to reduce the negative consequences on environment and health. Otherwise your paper will be purely focused on education.

Response:

Thank you. As suggested, we have added statements about disaster consequences on the environment and survivors’ health.

Please see page 1, lines 34-39 and page 9, lines 380-382 in the manuscript.

Reviewer 3 Report

Thank you for the opportunity to review the manuscript “An Exploration of Motivation for Disaster Engagement and Its Related Factors among Undergraduate Nursing Students in Taiwan”.

Congratulations to the authors for their work, I found your paper a potentially very valuable resource on Health Science and therefore an interesting and relevant contribution to IJERPH.

The manuscript explores the levels of and relationships between disaster nursing competence, anticipatory disaster stress, and the motivation for disaster engagement among undergraduate nursing students in Taiwan.

However, in my opinion there are several issues should be revised to improve the manuscript as noted below.

SPECIFIC COMMENTS:

TITTLE

Correct.

ABSTRACT

Correct.

INTRODUCTION

The introduction and background are very elaborate. The authors have defined all the concepts related to the object of the study.

METHOD AND RESULTS

The study suffers from several methodological problems:

  • The study was cross-sectional and analysed using correlations between disaster nursing competence, anticipatory disaster stress, and the motivation for disaster engagement. These were shown to be positive across anticipatory disaster stress, and the motivation for disaster engagement, however the association wes not strong, mainly below 0.5. Therefore, this will only account for approximately 25% of the variance. From these findings it is inferred that “Healthcare institutions and schools are suggested to work together to design disaster education plans which meet the goals of disaster nursing to improve students’ disaster competence and overcome personal stress when they encounter disaster events”. However, the results don´t showed a correlation between disaster nursing competence and anticipatory disaster stress. Thus, this is a major over-estimation of the findings and cannot be demonstrated.
  • A path analysis (multivariate analysis that could estimate the effect of multiple variables through structural equation models) can be used to model the associations between variables.
  • There is no reporting of reliability of the instruments used in this study.
  • Table 2 has an inappropriate format that makes it difficult to follow.
  • To compare mean scores between two-level categorical variables, the Student's t-test should be used. ANOVA must be performed when the categorical variable has more than three levels.

DISCUSSION AND CONCLUSION

The authors found statistical associations that were not strong and cannot infer causality in the conclusions. For this reason, the conclusion is a major over-estimation of the findings and cannot be demonstrated.

Author Response

Thank you for your review of our manuscript and your willingness to allow us to revise and resubmit the manuscript. Please see the attachment for our responses to your comments. We hope that we have addressed your major concerns.

Round 2

Reviewer 3 Report

It is curious that the authors justify that they cannot perform a path analysis (multivariate analysis that could estimate the effect of multiple variables through structural equation models) because the dependent variable (motivation for disaster engagement) was not normal distributed.
This raises two questions for me:

1) Why did you not use parameter estimation techniques to allow non-compliance with the assumption of multivariate normality? For example, weighted least squares (WLS), generalized least squares (GLS).
2) The second and most important question, and that affects the validity of the data analysis, is that if the dependent variable (motivation for disaster engagement) was not normal distributed, why do they use parametric techniques such as the ANOVA or the t-test to data analysis? The authors must justify this aspect, and if necessary, replace the analyzes with non-parametric techniques.

Author Response

It is curious that the authors justify that they cannot perform a path analysis (multivariate analysis that could estimate the effect of multiple variables through structural equation models) because the dependent variable (motivation for disaster engagement) was not normal distributed. This raises two questions for me:

Thank you for raising these important questions about statistical analysis. We have conducted analyses as suggested.

1) Why did you not use parameter estimation techniques to allow non-compliance with the assumption of multivariate normality? For example, weighted least squares (WLS), generalized least squares (GLS).

Response 1: We have conducted path analyses using GLS method. Thank you for your suggestion. Please see revisions on page 5, lines 240-246; page 7, lines 293-326.

2) The second and most important question, and that affects the validity of the data analysis, is that if the dependent variable (motivation for disaster engagement) was not normal distributed, why do they use parametric techniques such as the ANOVA or the t-test to data analysis? The authors must justify this aspect, and if necessary, replace the analyzes with non-parametric techniques.

Response 2: We have re-done the analyses using Mann-Whitney U test and Kruskal-Wallis test. Thank you for your correction. Please see revisions on page5, lines 235-237; page 7, lines 277-292.